# Hypertension screening, prevalence, treatment, and control at a large private hospital in Kampala, Uganda: A retrospective analysis

Usnish Majumdar[1,2], Rose Nanyonga Clarke[3], Andrew E. Moran[4], Patrick Doupe[5], Darinka D. Gadikota-Klumpers[6], Agaba Gidio[3], Dennis Ssentamu[3], David J. Heller[2]*

1 Department of Medicine, University of Pittsburgh Medical Center, Pittsburgh, PA, United States of America, 2 Arnhold Institute for Global Health, Icahn School of Medicine at Mount Sinai, New York, NY, United States of America, 3 Clarke International University, Kampala, Uganda, 4 Department of Medicine, Columbia University Vagelos College of Physicians and Surgeons, New York, NY, United States of America, 5 Zalando SE, Berlin, Germany, 6 Department of Medical Education, Icahn School of Medicine at Mount Sinai, New York, NY, United States of America

* david.heller@mssm.edu

**Data Availability Statement:** The de-identified dataset used for all analyses within this work is

## Abstract

Adult hypertension prevalence in Uganda is 27%, but only 8% are aware of their diagnosis, accordingly treatment and control levels are limited. The private sector provides at least half of care nationwide, but little is known about its effectiveness in hypertension control. We analyzed clinical data from 39 235 outpatient visits among 17 777 adult patients from July 2017 to August 2018 at Uganda's largest private hospital. We calculated blood pressure screening rate at every visit, and hypertension prevalence, medication treatment, and control rates among the 5 090 patients with two or more blood pressure checks who received any medications from the hospital's pharmacy. We defined hypertension in this group as 1) an average of two blood pressure measurements at separate consecutive visits, higher than 140 mm Hg systolic or 90 mm Hg diastolic, 2) receipt of any antihypertensive medication, or 3) the use of a hypertension electronic medical record code. We deemed hypertension control as normotensive at the most recent check. 12 821 (72.1%) of patients received at least 1 blood pressure check. Among the 5 090 patients above, 2 121 (41.6%) had hypertension (33.4% age-standardized to a world population standard): 1 915 (37.6%) with elevated blood pressure, and 170 (3.3%) were normotensive but receiving medication. 838 (39.4%) of patients with hypertension received medication at least once. Overall, 18.3% of patients achieved control (27% of treated patients, and 15% of untreated patients). Hypertension is common and incompletely controlled in this Ugandan private-sector population, suggesting several avenues for novel interventions.

available at https://figshare.com/s/85cb17f5871d34a0c361.

**Funding:** Research training for RNC was supported by the Fogarty International Center of the National Institutes of Health, and the U.S. President's Emergency Plan for AIDS Relief (PEPFAR); under a single award, number 1R25TW011213. The content in this manuscript is solely the responsibility of the authors and does not necessarily represent the official views of the National Institutes of Health. DJH reports receiving a research grant from Teva Pharmaceutical Industries. However, this grant was for work separate from this study and was not in any way involved in the conduct of this research. DJH also reports funding from Resolve, Incorporated (formerly Resolve to Save Lives), also for work separate from this study and in no way related to the conduct of this research. For all disclosed sources of funding above, the funders had no role in study design, data collection and analysis, decision to publish, or preparation of the manuscript.

**Competing interests:** I have read the journal's policy and the authors of this manuscript have the following competing interests: Dr. David J. Heller reports receiving funding form Teva Pharmaceutical Industries. However, this entity was not involved in any way in financially supporting this manuscript, nor any aspect of the data-gathering, analysis, content, or decision to publish.

## Introduction

Hypertension is among the leading risk factors for human mortality worldwide and is both more common and less well-controlled in many low-income countries, especially in sub-Saharan Africa, than in more affluent ones. Uganda, with one of the most rapidly-growing populations in the world, is no exception. Its estimated adult hypertension prevalence is now 28.9% in urban areas and 25.8% in rural areas, up from 14–18% as of 2005 [1]. Its rising prevalence is multifactorial, but sedentary lifestyles, an increasingly "Western" diet of simple carbohydrates and fats, and increasing use of alcohol and tobacco are key contributors [2, 3]. Hypertension and other non-communicable diseases cause 33% of all mortality in Uganda, but only 8% of persons with hypertension are aware of their diagnosis, and 3.6% achieve blood pressure control [4–6].

Fortunately, novel models for hypertension control in Uganda have begun to emerge. Some, for instance, have leveraged novel HIV/AIDS universal test-and-treat programs to screen patients for elevated blood pressure [7–9]. Others have leveraged nurses to screen for and treat hypertension and other NCDs as a result of physician shortages, a strategy called task-sharing [10, 11]. Years of experience have suggested how to prevent and remedy supply and medication shortages as well as gaps in patient data-tracking [12]. The majority of this work, however, has occurred in public-sector clinics, even though the private sector provides the majority of outpatient care for many Ugandans [13]. Recent work suggests that private-sector hypertension patients struggle to educate themselves on blood pressure control, while the physicians that treat them lack time or resources to aid their disease self-management, suggesting that nurse-led behavior change interventions for hypertension may be impactful in private as well as public contexts [10, 14]. However, little is known about the epidemiology of hypertension in the Ugandan private sector—or its baseline level of treatment or control. To address this gap, we analyzed clinical data from 39 235 outpatient visits among 17 777 patients at the largest private hospital in the nation.

## Methods

### Ethics statement

The protocol and data analysis plan for this study were approved by the Program for the Protection of Human Subjects (PPHS) at the Icahn School of Medicine at Mount Sinai, the Research Ethics Committee (REC) at the Clarke International University, and the Uganda National Council for Science and Technology. Because all data were de-identified prior to analysis, the requirement for informed consent of individuals in the data set was waived by each of these oversight boards.

### Design

We performed a retrospective observational study of electronic medical record (EMR) data and pharmacy invoicing data at a large private hospital in central Kampala, Uganda's capital. This private hospital system consists of a cluster of multi-specialty and primary care clinics located on site of a main hospital campus in Kampala which serves as the 'hub' of the health system, which additionally serves rural Ugandans through a network of smaller, rural primary care clinics. In this study, we evaluate the primary care-seeking population at the main hospital campus, the largest site of this health system. The data set did not capture the racial or ethnic background of participants, though the hospital serves a population that is chiefly African but also comprises minority South Asian and non-Hispanic white populations among others. Sex-specific results are as presented below.

### Study population and data set

The study population included all patients aged 18 or older seen at all outpatient primary care visits from July 2017 to August 2018. We included only visits to the hospital's general internal medicine clinic and family medicine clinic, because each such provider (unlike, for instance, an urgent care or surgical clinic) is expected to actively screen for, diagnose, and treat hypertension over the course of longitudinal (not one-off) care. We excluded all inpatient data, emergency department visits, as well as all other outpatient visits, including those at specialty clinics that would require referral from a general clinic. At every visit to the included clinics, hospital protocol requires blood pressure to be checked and recorded by a nurse in a sitting position prior to being seen by a doctor. The hospital uses the automated Edan M3A vital signs monitor for all blood pressure readings. By training protocol, blood pressure is checked once per visit, in either arm, with the patient in the seated position and the arm supported and at heart level. The hospital recorded all clinical data using the Navision EMR platform, a product of Microsoft Dynamics 365 Business Central. We used blood pressure data as recorded by that system, without direct observation of its measurement.

In addition to blood pressure values, we acquired visit characteristics (doctor type, insurance status), patient characteristics (age, sex) and all items procured at the hospital pharmacy (drugs and consumables). When documented, we also included diagnosis codes associated with each clinical encounter. These were missing in only 15.8% of visits overall, and 13.2% of persons with hypertension. All patients and doctor names were de-identified and represented by codes. The data set comprised 39 235 unique visits and 17 777 unique patients (**Fig 1**). We selected and cleaned the data for analysis via freely available Python packages [15].

### Outcomes

Our four main outcomes were rates of hypertension screening, prevalence, treatment with medication, and control. We defined a patient as *screened* for hypertension at a given visit if both systolic and diastolic blood pressure were documented in the Navision EMR. We defined the prevalence of hypertension via those who patients who met at least one of three criteria: 1) a documented blood pressure value of greater than or equal to 140 mm Hg systolic or 90 mm Hg diastolic on an average of any two consecutive checks (as per US Eighth Joint National Committee, or JNC-8, guidelines); 2) a documented diagnosis of hypertension (regardless of blood pressure value); or 3) documented procurement of one or more anti-hypertensive medications [see **S1 Table**] at the hospital pharmacy [16]. We did not include any upper or lower bounds on the duration of time between two visits to include in this analysis. We defined a patient as receiving *treatment* for hypertension who received any anti-hypertensive medication at the hospital pharmacy. Of note, this definition of treatment does not include non-pharmacologic measures that are not captured in this data–namely diet and lifestyle modifications. We defined a patient as achieving hypertension *control* whose most recent blood pressure value was under both 140 mm Hg systolic and 90 mm Hg diastolic, also per JNC-8 guidelines.

We calculated the rate of blood pressure screening for all visits—and therefore all patients—in the data set—a total of 39 235 visits among 17 777 patients. However, we calculated the prevalence of hypertension; the proportion of hypertension under treatment; and the proportion of hypertension under control only among patients who had 1) at least two separate blood pressure measurements and 2) at least one drug of any kind dispensed from the hospital pharmacy during the study period. The first prevalence criterion allowed us to focus only on patients whose hypertension status could be properly ascertained per JNC-8 guidelines. The second prevalence criterion allowed us to focus only on patients whose preferred pharmacy was at the hospital (as opposed to an outside facility) to accurately gauge who did and did not

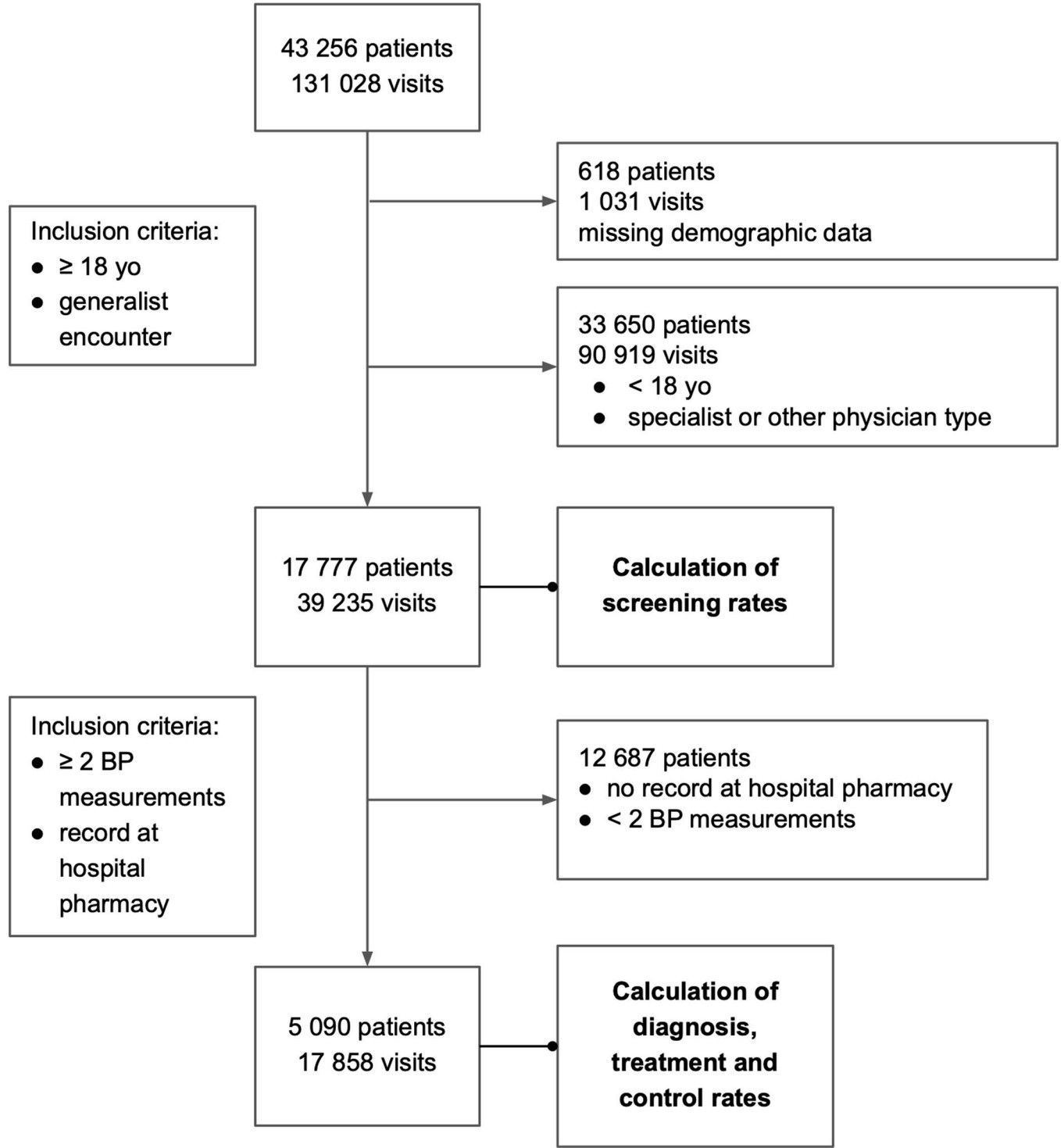

**Fig 1. Inclusion and exclusion flowchart.** All study members and their inclusion or exclusion from calculation of screening rates, as well as those included or excluded from calculation of prevalence, treatment, and control rates.

receive medication for hypertension. While the hospital pharmacy represents one of the largest pharmacies in the area, patients can also fill prescriptions at one of numerous local pharmacies. To our knowledge, many patients can and do pick up medications at both the hospital

pharmacy and their local pharmacies, as a matter of cost and convenience. Of note, only 5% (268) of patients with two BP measurements during the study period had not picked up a medication from the hospital pharmacy. These two criteria–two BP measurements and hospital pharmacy use—narrowed our sample size to 17 858 visits among 5 090 patients.

Because diagnosis strings appeared in the EMR via a free-text field (i.e. not a standardized set of codes), the data set referred to a hypertension diagnosis in multiple ways, such as "hypertension—essential", "hypertension", and "htn". We defined documented hypertension diagnosis for a given patient as a coded string containing the phrase "hypertension" or "htn", excluding strings that represent prehypertension ("prehypertension", "prehtn", etc.), intracranial hypertension, portal hypertension, and pulmonary hypertension appearing at any time during the study period. "High blood pressure" was not documented as hypertension, as this term was not found frequently in the data set and carries diagnostic ambiguity. Medications dispensed for hypertension at the hospital pharmacy were not free-text but selected from a list, so we used no such criteria—rather, we identified 48 medications for hypertension treatment [see **S1 Table**] and defined a patient as under hypertension treatment (and therefore experiencing hypertension) who received such a medication at the hospital pharmacy at any time during the study period. To generate this list of medications, we classified all dispensed agents in the dataset according to their Anatomical Therapeutic Chemical (ATC) category and subcategory per World Health Organization criteria. This approach yielded 13 distinct categories and combinations of medications: calcium channel blockers (CCBs); beta blockers; thiazide diuretics; angiotensin-converting enzyme inhibitors (ACE-Is); angiotensin receptor blockers (ARBs); potassium-sparing diuretics; loop diuretics; peripheral alpha-adrenoreceptor antagonists; central alpha-adrenoceptor agonists; combination alpha-beta blockers, arteriolar smooth muscle relaxants; combination ARB-thiazide diuretics; and ARB-CCB combination agents. We presumed that the indication for any use of any such agent in the data set was the treatment of hypertension. Lastly, we age-standardized any hypertension rate to the WHO 2000–2025 World standard population.

## Statistical analysis

In order to compare both the study sample (17 777 patients) and the subset of this sample chosen for evaluation of prevalence, treatment, and control (5 090), we performed a Cramer's V-test (derived from the $\chi^2$ statistic but more suitable for large sample sizes) on the distribution of ages among selected ranges (18–30, 30–45, 45–65, and 65+), sex, and insurance status.

We examined potential associated factors via multiple logistic regression of each of the four main outcome variables: screening (at visit level), prevalence (at patient level); treatment (at patient level); and blood pressure control (at patient level). Age, sex, and the total number of outpatient generalist visits during the study period were included as independent variables. We also included payment type as an independent variable: because all hospital patients lacking insurance must pay in cash (self-pay), we treated payment type as a binary variable comprising either cash or insurance. Patients who used both methods of payment during the study period were assigned the payment type they used most frequently. For the regression examining blood pressure control as an outcome variable, we additionally included blood pressure treatment as an independent variable.

We generated odds ratios and 95% confidence intervals for each analysis, holding a p value < 0.05 to be statistically significant. All regressions and statistical tests were performed in Python by utilizing the *statsmodels* package, an econometric and statistical modeling toolkit, as well as other freely available packages such as pandas, a general-purpose data management library [17, 18].

## Results

### Baseline characteristics

The screening sample consisted of 39 235 visits and 17 777 patients ranging from 18 to 97 years old (mean: 36.8), whereas the subset of these patients chosen to analyze prevalence, treatment, and control consisted of 17 858 visits and 5 090 patients ranging from 18 to 88 years old (mean: 37.7). More than half the original screening sample are described by diagnosis field as "general checkup" of which the leading diagnoses were diabetes, heart failure, angina, and hypertension. Among these 5 090 patients, the mean number of visits during the study period was 4.25 (minimum 2, median 3, $25^{th}$ percentile 2, $75^{th}$ percentile 5, maximum 32). The median interval between consecutive visits was 29 days (mean 59 days, $25^{th}$ percentile 6, $75^{th}$ percentile 91 days, max 399 days–the duration of the study period). The distribution of sex, age bins, and insurance status are not significantly different between the original study population and sample (Cramer's V test, p = 0.99 [age], 0.87 [sex], 0.62 [insurance status]), as can be seen in **Table 1** and **Fig 2**. See **S1 Fig** for a Venn diagram comparing the different hypertension criteria included in the evaluation of prevalence.

### Screening rates

Among 39 235 eligible visits, 25 352 (65%) recorded both a systolic and diastolic blood pressure. Among 17 777 eligible patients, 12,821 (72.1%) of patients received at least 1 complete blood pressure measurement during the study period (**Table 2**). In a multiple logistic regression predicting screening with age, gender, number of visits, and insurance status, the male gender was associated with decreased odds of screening (OR 0.86, 95% CI 0.80–0.92, p < 0.001). Number of visits (OR 1.78, 95% CI 1.71–1.86, p < 0.001) and insurance status (OR 2.73, 95% CI 2.54–2.92, p < 0.001), were associated with increased odds of screening, as shown in **Fig 3**.

### Prevalence

Among 5 090 patients in the prevalence and treatment sample (who had at least 2 blood pressure measurements and at least 1 hospital pharmacy record during the study period), 2 121 (42%) had hypertension per EMR code, receipt of antihypertensive therapy, or by blood

**Table 1. Baseline characteristics of study population and sample.**

| | Screening Cohort | | Prevalence & Treatment Cohort | | |
|---|---|---|---|---|---|
| | (n = 17 777) | | (n = 5 090) | | p (Cramer's V test) |
| **Age Quartile** | | | | | *0.99* |
| 18–30 | 6 453 | (36%) | 1 648 | (32%) | |
| 30–45 | 7 548 | (42%) | 2 172 | (43%) | |
| 45–65 | 3 295 | (19%) | 1 139 | (22%) | |
| 65+ | 481 | (3%) | 131 | (3%) | |
| **Sex** | | | | | *0.87* |
| Male | 8 610 | (48%) | 2 351 | (46%) | |
| Female | 9 167 | (52%) | 2 739 | (54%) | |
| **Insurance Status** | | | | | *0.62* |
| Self-Pay | 7 126 | (40%) | 955 | (19%) | |
| Insured | 10 651 | (60%) | 4 135 | (81%) | |

All p-values resulted from Cramer's V test performed on Age Quartile, Sex, and Insurance Status distributions between the two cohorts.

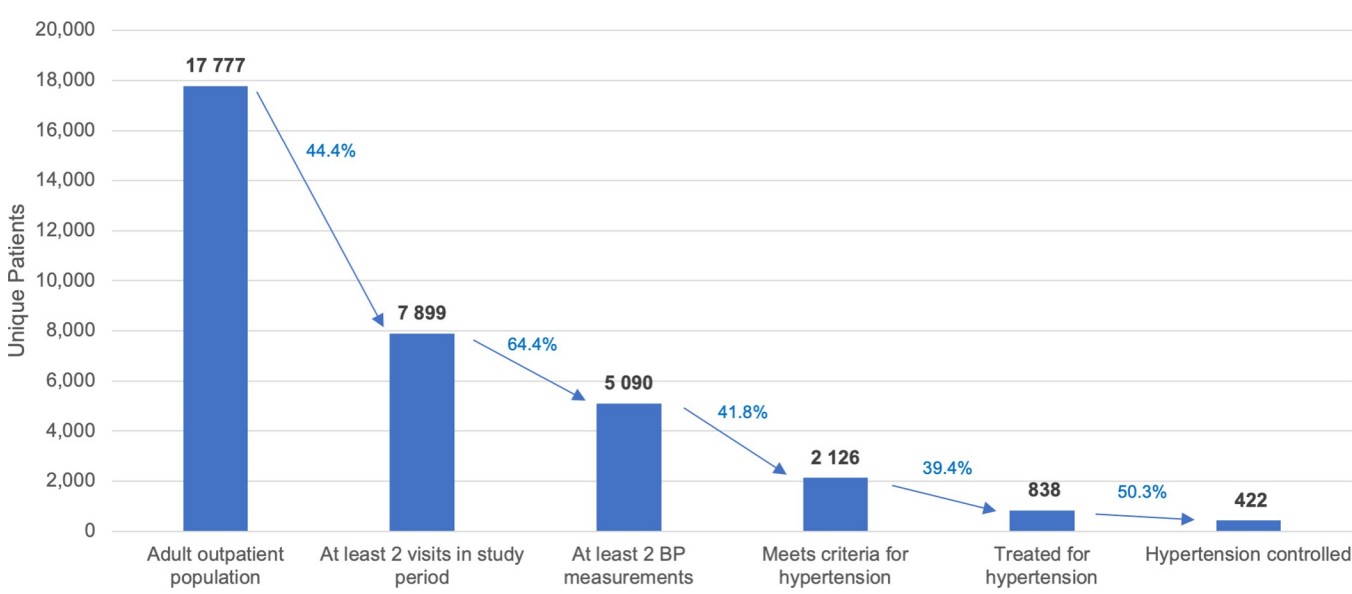

**Fig 2. Cascade of care.** Bars in graph depict extent of correct, and imperfect, treatment and control across steps of care cascade.

pressure measurements (**Table 2**). The age-standardized prevalence of hypertension in this sample was 33.4%. Among 2 121 patients, 1 915 met hypertension criteria by blood pressure measurements, 838 by receipt of antihypertensive therapy, and 668 by EMR code. See **S1 Fig** for a Venn diagram comparing the different hypertension criteria included in the evaluation of prevalence. Of the 2 085 patients who had hypertension by blood pressure or receipt of antihypertensive medications, 632 (30%) were diagnosed as hypertensive by EMR code. 36 patients had a 'hypertension' EMR code alone.

Overall, older patients had higher rates of hypertension prevalence than younger patients (79% in 66+, as compared to 23% among ages 18–30), as did male patients (46%) when compared to female patients (38%). In a multiple logistic regression describing hypertension prevalence and its correlates, older age (OR 1.06, 95% CI 1.05–1.06, p < 0.001), male gender (OR 1.15, 95% CI 1.04–1.28, p = 0.007), number of visits (OR 1.13, 95% CI 1.11–1.14, p < 0.001),

**Table 2. Overview of screening, prevalence, and treatment.**

| | | Screened | | Prevalence | | Treatment | | Control, Treated | | Control, Untreated | |
|---|---|---|---|---|---|---|---|---|---|---|---|
| | n = | 17 777 | | 5 090 | | 2 121 | | 838 | | 1 288 | |
| **Overall** | | 12 821 | (72%) | 2 121 | (42%) | 838 | (39%) | 223 | (27%) | 199 | (15%) |
| **Age** | | | | | | | | | | | |
| 18–30 | 6 453 | 4 605 | (71%) | 381 | (23%) | 47 | (12%) | 14 | (30%) | 44 | (13%) |
| 31–45 | 7 548 | 5 513 | (73%) | 869 | (40%) | 290 | (33%) | 83 | (29%) | 93 | (16%) |
| 46–65 | 3 295 | 2 391 | (73%) | 767 | (68%) | 433 | (56%) | 107 | (25%) | 53 | (16%) |
| 66+ | 481 | 312 | (65%) | 104 | (79%) | 68 | (65%) | 19 | (28%) | 9 | (25%) |
| **Sex** | | | | | | | | | | | |
| Male | 8 610 | 6 051 | (70%) | 1 075 | (46%) | 432 | (40%) | 99 | (23%) | 103 | (15%) |
| Female | 9 167 | 6 770 | (74%) | 1 046 | (38%) | 406 | (39%) | 124 | (31%) | 96 | (16%) |
| **Payment Type** | | | | | | | | | | | |
| Insurance | 10 651 | 8 491 | (80%) | 1 830 | (42%) | 729 | (40%) | 190 | (26%) | 154 | (14%) |
| Cash | 7 126 | 4 330 | (61%) | 291 | (40%) | 109 | (34%) | 33 | (30%) | 45 | (22%) |

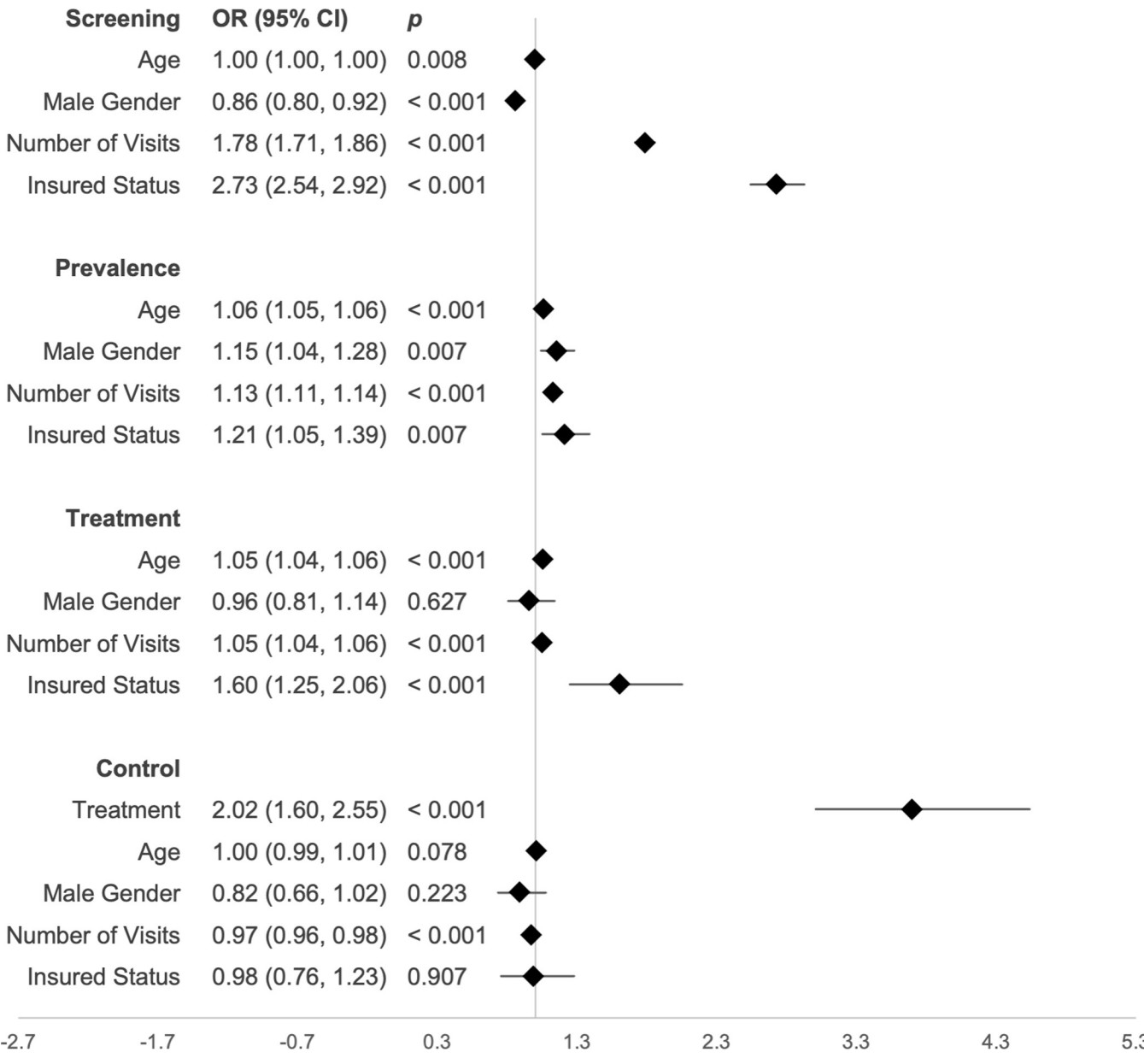

**Fig 3. Multiple logistic regressions.** Each multivariate logistic regression model predicts the likelihood of an individual being screened for hypertension, having hypertension, treated for hypertension, or controlled.

and insured status (OR 1.21, 95% CI 1.05–1.39, p = 0.007) were associated with increased likelihood of hypertension, as shown in **Fig 3**.

### Treatment (Medication)

Of 2 121 patients with hypertension per study criteria, 838 (39.5%) received antihypertensive medication from the hospital pharmacy during the study period (**Table 2**). The 646 hypertensive patients identified by diagnosis code had a higher rate of medication treatment (83.4%) than those meeting criteria for hypertensive but without a diagnosis code (37.9%). Overall, 838 patients received 4 776 itemized prescriptions throughout the study period. Calcium-channel

**Table 3. Prescription data.**

| Antihypertensive Category | Number of prescriptions | | Most common drug in category |
|---|---|---|---|
| CCB | 1 916 | (40%) | amlodipine 10mg |
| ARB + Thiazide Diuretic | 1 465 | (31%) | losartan 50mg + hydrochlorothiazide 12.5mg |
| Beta Blockers | 552 | (12%) | bisprolol 5mg |
| ARB | 339 | (7%) | losartan 50mg |
| Loop Diuretic | 95 | (2%) | furosemide 40mg |
| Thiazide Diuretic | 94 | (2%) | bendrofluazide 5mg |
| Alpha + Beta Blockers | 79 | (1.6%) | carvedilol 6.25mg |
| ARB + CCB | 78 | (1.6%) | losartan 50mg + amlodipine 5mg |
| ACE Inhibitor | 57 | (1.2%) | enalapril 5mg |
| K- sparing Diuretic | 47 | (1.0%) | spironolactone 25mg |
| Arteriolar smooth muscle relaxants | 41 | (0.9%) | hydralazine 25mg |
| Alpha- and Beta Blockers | 9 | (0.2%) | labetolol 100mg |
| Central Alpha-adrenoceptor agonist | 2 | (0.04%) | clonidine 10mcg |
| Peripheral | 2 | (0.04%) | alfuzosine 10mg |

blockers (1 916 prescriptions, 40%) and angiotensin-receptor blocker / thiazide diuretic combination pills (1 465 prescriptions, 31%) comprised the majority of medications prescribed (**Table 3**). In a multiple logistic regression describing medication treatment of hypertension, older age (OR 1.05, 95% CI 1.04–1.06, p < 0.0001), number of visits (OR 1.05, 95% CI 1.04–1.06, p < 0.0001), and insured status (OR 1.60, 95% CI 1.25–2.06, p < 0.0001) were associated with increased likelihood of hypertension treatment (**Fig 3**).

## Control

Among 2 121 patients with hypertension, 422 (19.5%) were controlled (normotensive at their last visit). Rates of control were similar across age and gender, but higher in the medication-treated group (27%) than in the untreated group (15%) (**Table 2**). Of 1 915 patients meeting criteria by clinical measurements alone, those who had a baseline SBP of >160mm Hg systolic had a lower rate of control (8.4%) than those with a baseline SBP 140-160mm Hg (12.4%). In a multiple logistic regression describing control of hypertension, antihypertensive treatment (OR 2.02, 95% CI 1.60–2.55, p < 0.001) was associated with increased likelihood of treatment control, whereas number of visits (OR 0.97, 95% CI 0.96–0.98, p = 0.01) and insured status (OR 0.68, 95% CI 0.51–0.91, p = 0.01) were associated with decreased likelihood of treatment control.

## Discussion

Our analysis of one year of blood pressure data in this large private health system sample—in an under-studied and low-resource setting—reveals a high prevalence of hypertension (42%, age-standardized to 33%) relative to those reported in high-income countries (25.3–31.6%), low-middle income countries (age-standardized 31.2–31.7%), and prior estimates in urban Uganda (28.9%) [5, 19]. Rates of treatment with medication (39.4%) and control (20% overall; 27% if medicines and 15% without) were also significantly less than in high-resource settings such as the US or OECD countries [19, 20]. Compared to low- and middle-income countries overall, treatment rate in this cohort was significantly greater (39.4%) than reported averages (29.0%) [19]. However, control rates were lower (19.5%) than comparable populations in sub-Saharan Africa (22.6%)—yet greater than documented in previous work in Uganda (9.4%) [21, 22]. These outcomes were particularly favorable relative to rural Uganda, where over 90% of

hypertension is undiagnosed and under 5% of those diagnosed achieve control [5, 7]. In any case, insofar as the "cascade" of hypertension care causes cumulative attrition at each stage, there exist significant opportunities for this private hospital setting to improve its hypertension management strategy at a population level [9]. If those 4 956 patients unscreened for hypertension (no blood pressure measurements at visit) have the same 42% prevalence rate as the study sample (screened twice, picked up medications at hospital pharmacy), some 2 081 persons may have undiagnosed hypertension undetected by our approach.

This private Ugandan hospital, in which approximately 60% of patients have access to health insurance and others pay in cash, likely serves a significantly higher socioeconomic status population than the Ugandan public sector, yet concurrently a lower-income population than most US hospitals (given the income disparity between even the highest-income Ugandan households and the lowest-income US ones). Its control outcomes for hypertension control between the Ugandan and US population averages may therefore reflect how starkly resource access disparities drive health outcomes: an intermediate level of patient and health system affluence between those two poles yields intermediate clinical results [23, 24]. A prior study of the relationship between individual income level and hypertension control found that, in urban clinics among lower-income Sub-Saharan countries, lower individual wealth was associated with both lower likelihood of control and higher grade of hypertension [22]. In the US, those with insurance have nearly twice the rate of hypertension control (43–54%) than those without (24%) [25].

Relative to other hypertension control studies in Uganda and the East African region, our facility-level findings demonstrate higher levels of blood pressure treatment and control than in the general population, yet also offer insight into how routine primary care operates–outside of a pilot study or other research setting. For example, the Sustainable East African Research in Community Health (SEARCH) study achieved 75% medication treatment and 46% blood pressure control among 3,380 persons linked to hypertension care in Uganda and Kenya–significantly greater than observed in this population–but in a program not yet integrated within routine care [8]. Similarly, the Linkage And Retention to hypertension care in rural Kenya (LARK) achieved a mean 13 mm systolic blood pressure decrease in persons with hypertension treated with smartphone behavior counseling and medication, with 26% control rate, in an intervention now being scaled and integrated with microfinance support, but not yet a part of routine care in western Kenya [9, 26, 27]. Given that the private sector provides more than half of health care in Uganda, and given than hypertension care cascade analyses in both private and public sector clinics in East Africa are uncommon, our work provides novel insights into both the baseline state of usual care in a higher-resource Ugandan clinic, and thereby opportunities to strengthen it through improvements in medication treatment and adherence–work now underway at this site [13, 14, 28–31].

The significant care disparities between insured (those with better access to healthcare) and uninsured patients within the hospital cohort further supports this hypothesis. Among persons meeting study criteria for hypertension, 34% percent of the uninsured received medication treatment, compared to 40% percent of the insured. After correcting for medication treatment status, insurance status was no longer correlated with hypertension control, suggesting that differential receipt of medications between these two groups explains the observed control disparity. Because of limitations in our study approach (which documented as treated only those who picked up a medicine at the hospital pharmacy, regardless of what was prescribed), we cannot establish to what extent insured persons were more likely to be offered medicine by their physician, as opposed to more likely (and more financially able) to access it. It is suggestive, however, that in the population with enough BP measurements and hospital pharmacy records, 81% were insured versus 60% in the general screening population. Further research

should seek to better understand the barriers to medication treatment for patients with hypertension at this and similar hospitals in low-income countries, be it inability to afford medication, therapeutic inertia, reluctance to take medication, or other causes.

Regardless of treatment disparities, however, 27% of those treated for hypertension with medication achieved blood pressure control. This result, which also compares negatively with high-resource settings (in the US, 59.5%), reinforces prior data from high-income settings that access to hypertensive medication (widely known to be an essential therapy but nonetheless widely unavailable in Uganda)- is nonetheless insufficient to achieve blood pressure control on its own [32, 33]. The gap in control rates between *treated* patients in this cohort and *treated* patients in high-income settings suggests further disparities: in access to the support required to adhere to antihypertensive medicines, as well as in access to counseling treatment around lifestyle and diet that complements antihypertensive pharmacotherapy treatment.

This finding suggests a separate opportunity for further research: into the extent, barriers, and potential solutions that address medication adherence for hypertension and other non-communicable diseases in Uganda and similar settings. Although medication nonadherence for chronic disease is under extensive study in the United States, data are more sparse in Uganda [7, 9, 34, 35]. Previous work on hypertension control barriers in Uganda details frequent and unpredictable medication stockouts, and limited evidence suggests that inconsistent access to medications may drive loss to follow-up [12, 36]. Nonetheless, it remains unclear whether inconsistent medication access impairs medication adherence relative to an uninterrupted medication supply, as most prior work on medication adherence in Uganda has occurred only in the setting of inconsistent medication access.

Hypertension medication adherence research in hospital cohorts such as this—where a patient's receipt or non-receipt of medicine at the hospital pharmacy can be confirmed, and where a sizable cohort of patients already access medicine consistently—could address this question. Previous work in this hospital system has demonstrated that a hypertension "adherence club" resulted in significant blood pressure control relative to baseline and relative to the results above, but this study lacked a control group and the effect of medication adherence counseling (as opposed to behavior counseling of other types) remains unclear [10]. Qualitative studies in this setting further reinforced patient and provider interest in hypertension education and adherence counseling [14, 28]. Work in this setting of relative medication abundance could inform hypertension behavior counseling in the Ugandan public sector, where adherence research remains hampered by frequent stockouts but hypertension remains common and public education limited [14, 24, 28, 37, 38].

Our work has several notable limitations, many of which stem from our use of routine care data in this study. Although we used one year of retrospective cohort data, we analyzed the key outcomes of screening, prevalence, and medication treatment cross-sectionally, i.e., we considered a subject to meet these three criteria for the entire study period if they achieved it at any point during the year. We therefore do not make causal inferences regarding how and why these variables correlate. For example, insurance status may correlate with hypertension status due to underlying greater prevalence of hypertension in the insured; or conversely a greater proclivity for those with hypertension to seek insurance. Moreover, these data rely on inference and may not fully reflect underlying behaviors or outcomes: for instance, hypertension may not be less common in the uninsured but merely less often recorded by the physician—due to less frequent blood pressure checks, but perhaps also less diagnostic documentation or medication treatment. Similarly, the uninsured may not in fact be less often prescribed medication for hypertension than the insured, but merely more likely to purchase hypertension medication at a non-hospital pharmacy (and therefore be labeled as 'untreated' in our analysis). We worked to mitigate these limitations, however, by adjusting for measured confounders

(i.e. treatment as a confounder of the relationship between insurance and control) and also limiting our analyses to those known to use the hospital pharmacy.

Another limitation stems from not placing upper or lower bounds on the interval between consecutive blood pressure measurements considered for hypertension prevalence–visits too closely spaced may limit the accuracy of our approach. Fortunately, visits only one or two days apart were uncommon (median interval 29 days, 25th percentile 6 days), and likely had a minimal effect on study findings. The short study duration (just over 1 year) is also another potential limitation in this study–given the large proportion of patients who were screened for hypertension but only visited the clinic once during the study period, extending the study period may well have captured cases of hypertension missed during the current period. Limiting our analysis to those known use to the hospital pharmacy may limit the external validity of this study, as it is not well known what causes a patient to seek medications at the hospital pharmacy versus elsewhere. Additionally, should a patient who otherwise picked up medications at the hospital pharmacy choose to fill a prescription at a local pharmacy, this would have not been captured in our data and led to an underestimate of treatment.

Treatment of hypertension had a narrow definition in this study–namely, pharmacologic treatment by one of the 48 antihypertensive agents. Notably, this does not include the many non-pharmacologic approaches to hypertension management, including myriad diet and lifestyle interventions. This may additionally explain the relatively weak association between treatment and control (OR close to 2), Separately, any of these antihypertensive agents carry secondary indications for non-hypertensive disorders–for example, spironolactone functions not only as an antihypertensive but also as a mainstay of treatment in some cirrhosis and heart failure patients. Our conservative approach in this setting–including all medications with any indication in hypertension–may result in an overestimate of treatment for hypertension. In describing prevalence, treatment, and control, we utilized age and number of visits as continuous variables in our logistic regression models–one potential limitation of this approach is that the effects of these variables may not be linear.

We measured hypertension control, unlike the other variables above, based only on the most recent blood pressure check conducted. This approach, common to hypertension research, allowed us to some extent to use a retrospective cohort approach, as patients typically underwent hypertension treatment, or obtained multiple visits for hypertension, before that blood pressure check at the end of the study period. However, it remains unclear whether the number of blood pressure visits correlates negatively with control due to (1) prior lack of control prompting more visits or (2) the frequent visits negatively influencing the control outcome. The natural physiologic variability in blood pressure is a further limitation: the most recent single documented blood pressure is not always reflective of underlying hypertensive control, just as the two or more documented blood pressures meeting our criteria for hypertension over the study year may overlook those who met such criteria during years prior but not documented or treated. Furthermore, we were unable to confirm whether the patient was correctly positioned or the cuff properly sized and applied to optimally measure blood pressure. Nonetheless, even after setting the hypertension threshold at a conservatively high level of 140 mm Hg systolic or 90 mm diastolic to improve specificity in our prevalence and control data, we found both a high hypertension prevalence and a large fraction of those whose pressure remained elevated despite treatment with medication.

## Conclusions

This study provides novel data on hypertension screening, prevalence, and pharmacotherapy treatment in a leading private-sector Ugandan hospital with a large on-site pharmacy. Little

data has been reported on hypertension in the private sector in sub-Saharan Africa, and here we report high hypertension prevalence with medication treatment and control rates worse than in high-income countries but superior to rural Ugandan districts. Our findings suggest significant barriers to hypertension control in Uganda even in a setting where pharmacotherapy is nominally universally available. Further avenues for research include identifying and addressing barriers to medication adherence (if not access) and behavior change. Successful care models could benefit the Ugandan public sector and other low-resource settings that face similar challenges.

## Supporting information

**S1 Table. List of medications considered to be antihypertensive therapies in this study population.**
(DOCX)

**S1 Fig. Venn diagram of hypertension criteria.**
(TIFF)

## Acknowledgments

We gratefully acknowledge the patients, providers, and staff at the hospital where this research was conducted, and whose support made this work possible. We further thank the research support teams at Clarke International University and the Arnhold Institute for Global Health whose assistance supported this research.

## Author Contributions

**Conceptualization:** Usnish Majumdar, Rose Nanyonga Clarke, Andrew E. Moran, Patrick Doupe, Darinka D. Gadikota-Klumpers, Agaba Gidio, David J. Heller.

**Data curation:** Patrick Doupe, Dennis Ssentamu, David J. Heller.

**Formal analysis:** Usnish Majumdar, Andrew E. Moran, Patrick Doupe, Darinka D. Gadikota-Klumpers, David J. Heller.

**Funding acquisition:** David J. Heller.

**Investigation:** Usnish Majumdar, Rose Nanyonga Clarke, Andrew E. Moran, Darinka D. Gadikota-Klumpers, Agaba Gidio, David J. Heller.

**Methodology:** Usnish Majumdar, Rose Nanyonga Clarke, Andrew E. Moran, Agaba Gidio, David J. Heller.

**Project administration:** Rose Nanyonga Clarke, Dennis Ssentamu, David J. Heller.

**Resources:** Rose Nanyonga Clarke, Dennis Ssentamu, David J. Heller.

**Supervision:** Rose Nanyonga Clarke, Andrew E. Moran, Patrick Doupe, Agaba Gidio, Dennis Ssentamu, David J. Heller.

**Validation:** Andrew E. Moran, Agaba Gidio.

**Writing – original draft:** Usnish Majumdar.

**Writing – review & editing:** Usnish Majumdar, Rose Nanyonga Clarke, Andrew E. Moran, Patrick Doupe, Darinka D. Gadikota-Klumpers, Agaba Gidio, Dennis Ssentamu, David J. Heller.

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
