## [Decision Letter · Decision Letter 0]

6 Oct 2021

PGPH-D-21-00519

Hypertension screening, diagnosis, treatment, and control at a large private hospital in Kampala, Uganda: a retrospective analysis

Dear Dr. Heller,

Thank you for submitting your manuscript to PLOS Global Public Health. After careful consideration, we feel that it has merit but does not fully meet PLOS Global Public Health’s publication criteria as it currently stands. Therefore, we invite you to submit a revised version of the manuscript that addresses the points raised during the review process.

We look forward to receiving your revised manuscript.

Kind regards,

Andre F. S. Amaral, Ph.D.

Academic Editor

Journal Requirements:

1. Please provide separate figure files in .tif or .eps format only.

2. Please update the completed 'Competing Interests' statement, including any COIs declared by your co-authors. If you have no competing interests to declare, please state "The authors have declared that no competing interests exist". Otherwise please declare all competing interests beginning with the statement "I have read the journal's policy and the authors of this manuscript have the following competing interests:

3. In the online submission form, you indicated that your data will be submitted to a repository upon acceptance.  We strongly recommend all authors deposit their data before acceptance, as the process can be lengthy and hold up publication timelines. Please note that, though access restrictions are acceptable now, your entire data will need to be made freely accessible if your manuscript is accepted for publication. This policy applies to all data except where public deposition would breach compliance with the protocol approved by your research ethics board. If you are unable to adhere to our open data policy, please kindly revise your statement to explain your reasoning and we will seek the editor's input on an exemption. Please be assured that, once you have provided your new statement, the assessment of your exemption will not hold up the peer review process.

4. Please amend your Financial Disclosure statement. This is published with the article, therefore should be completed in full sentences and contain the exact wording you wish to be published.

i) State the initials, alongside each funding source, of each author to receive each grant.

ii). State what role the funders took in the study. If the funders had no role in your study, please state: “The funders had no role in study design, data collection and analysis, decision to publish, or preparation of the manuscript.”

Additional Editor Comments (if provided):

Please pay extra attention to the major issues raised by the reviewers. In addition to their comments, consider improving the discussion of the study limitations, particularly considering its design, setting, population and external validity.

Reviewers' comments:

Reviewer's Responses to Questions

**Comments to the Author**

1. Does this manuscript meet PLOS Global Public Health’s publication criteria? Is the manuscript technically sound, and do the data support the conclusions? The manuscript must describe methodologically and ethically rigorous research with conclusions that are appropriately drawn based on the data presented.

Reviewer #1: Yes

Reviewer #2: Partly

2. Has the statistical analysis been performed appropriately and rigorously?

Reviewer #1: Yes

Reviewer #2: No

3. Have the authors made all data underlying the findings in their manuscript fully available (please refer to the Data Availability Statement at the start of the manuscript PDF file)?

Reviewer #1: No

Reviewer #2: No

4. Is the manuscript presented in an intelligible fashion and written in standard English?

Reviewer #1: Yes

Reviewer #2: Yes

5. Review Comments to the Author

Reviewer #1: Usnish Majumdar and colleagues present an analysis of hypertension screening, diagnosis, treatment and control of electronic medical record data of over 17500 internal medicine and cardiology outpatients at the largest private hospital in Uganda. The data are well analysed and presented and add important information on hypertension care from the private sector to available information from the public sector. As the analysis is restricted to internal medicine and cardiology outpatients of the hospital, this should be reflected in the title and abstract of the paper.

The limited data available for analysis – blood pressure values, visit characteristics (doctor type, insurance status), patient characteristics (age, sex) and items procured at the hospital pharmacy limit a more comprehensive comparison with public sector patients and do not allow to assess determinants for screening, diagnosis, treatment and control of hypertension, providing insight in who is most at risk of not being screened, diagnosed, treated and well controlled. It is this information that would be most useful for in-hospital quality audit of services rendered to optimize patient care and to allow for more informative comparison with hypertension care in public facilities for practice and policy. While this may be difficult to address in the current paper, future use of EMR data should advocate for more comprehensive data for analysis.

The paper would significantly benefit from a more detailed comparison of hypertension care with previous studies undertaken in the country and elsewhere in the region, respectively sub-Saharan Africa (e.g. Bay et al., BMC Health Serv Res 2019; Sorato et al., BMC Cardiovasc Disord 2021) and LMIC in general (e.g. Geldsetzer et al. Lancet 2019 ; Sudharsanan et al., Circulation 2021) and between private and public facilities.

Reviewer #2: PGPH-D-21-00519

Plos Global Public Health

The importance of the study subject in the study setting cannot be overemphasized.

Hypertension remains one of the most important determinants of death and disability worldwide, with its impact increasing in LMIC, particularly in sub-Saharan Africa, and it is difficult to accept and worth studying why all we know so far is not translating into population benefit.

I congratulate the authors for addressing this issue, making use of secondary data, namely routine care data, to do this study.

However, I have some serious concerns regarding bias of several types. Part of the limitations I will describe now could be overcome or at least reduced with a different analytic approach, with the same data. Some others may be impossible to overcome with this data. The most important concerns are acknowledged by the authors in the discussion, but the problem is that they are too serious to become acceptable with a simple mention in the limitations section.

Specific comments (in the order of the text, not ordered by importance):

1. It is strange to have a population approach and a descriptive objective regarding prevalence of hypertension in the setting of one hospital. The authors should at least describe the system better, explain what they mean by primary care at hospital, and describe the population the hospital serves, for what purposes, how the patients get there (referral), etc. [title, line 297, and more]

2. Line 32 – the word “efficacy” is not applicable here. Effectiveness, at best.

3. Line 37 – two or more blood pressure checks regardless of the time interval between them? This could result in someone with only 2 blood pressure measurements separated by almost 1 year being misclassified according to the JNC principles. I suggest considering only 2 consecutive visits separated by no more than, say, 1 or 2 months.

4. Lines 38-39 – “an average of two blood pressures at separate…” should read “an average of two blood pressure measurements at separate…”

5. Line 41 – what is the coding system/classification used?

6. Line 44 – age-standardised prevalence is a meaningless value unless the standard is known, for comparison purposes. I know it is in the text, but the abstract must be understandable alone.

7. Line 94 – “the term “hypertension specialty” is not clear and it doesn’t refer to a specific concept. I suggest “specialty visits considered most likely to xxxx”. I leave to the authors completing this sentence.

8. Line 94 – When the authors write “We included only visits to the hospital’s general internal medicine clinic and cardiology” do they mean beyond primary care (as written in the previous sentence) or truly only those?

9. Internal medicine and cardiology are the specialties where patients with hypertension are likely referred to due to their hypertension. This population is not appropriate at all to have a population approach to hypertension prevalence or screening. THIS IS A MAJOR POINT.

10. Lines 100 and 103 contradict themselves – is the hospital protocol to measure BP in the non-dominant arm or in either arm?

11. Line 110 – “when documented we also included diagnosis codes”. It is very important to describe how many visits (what proportion of visits) have at least one code documented.

12. CONSORT is for randomized controlled trials, not applicable to this study.

13. Line 121 and general question in the article – the outcome that the authors call “hypertension diagnosis” is in fact “hypertension prevalence”, defined as fulfilling criteria for diagnosis, which are then specified. Many of those patients were not exactly diagnosed. THIS IS A MAJOR POINT. The whole text, tables and figures need to be revised accordingly. In line 318 the authors themselves correctly say “persons meeting diagnostic criteria for hypertension”.

14. Lines 129-130 – it would be very important and useful to describe the system regarding access to medicines, supplies, distribution and patterns of use. Where else can patients get their medications? Are medical prescriptions necessary? How many patients who get one or more drugs at the hospital pharmacy can also get others during the year at other selling sites? THIS IS A MAJOR POINT.

15. Line 131 – the most recent blood pressure value could be the second value that was considered for hypertension definition, in patients with only 2 visits or 2 BP measurements. This is unacceptable to define control. I suggest control should be considered only among treated patients and based on BP measurement AFTER treatment. Also, if this BP measurement is made too soon after treatment initiation (due to the study stop at 1 year) it may be inadequate to define control when “stable”, which takes several weeks.

16. Line 138 – considering the inclusion criterion of having at least one drug of any kind dispensed from the hospital pharmacy in one year seems to me to be too restrictive; I suppose much of this population may not get any drug either at this hospital or at any other place in a 1-year period, even if having medical appointments. Therefore, I suggest that this criterion should be applied only to define treatment and control (for the reason presented by the authors, which I understand) but not for hypertension prevalence, to avoid reducing representativeness so much.

17. Lines 145-149 – What if in the EMR it was written “high blood pressure” or “hypertensive”? On the other hand, how did the authors disregard other forms of hypertension such as intracranial hypertension, portal hypertension, pneumothorax, etc?

18. Lines 150-152 – if the medications at the pharmacy are, fortunately, coded, don’t describe the data extraction as not having to use free text searches, but simply describe it the positive way, as being selected from a list.

19. Lines 162-163 – I know it is difficult to do it in any other way (having the drug and the indication), particularly if not even the diagnoses are registered. However, you should at least discuss what other indications could be at stake, and if their prevalence is high or low in this population. Also, loop diuretics are probably not used for hypertension but rather heart failure, isn’t this right?

20. Line 174 – by estimating screening rates at the visit level, did the authors take into account the interval between visits? For example, in a follow up visit very shortly after a previous visit (for example the next day, for the same reason, unrelated to hypertension) it would be reasonable not to measure BP again.

21. Line 193 – it would be informative and important to describe the reasons/diagnoses in the 39235 visits.

22. Line 195 – please describe the distribution of the number of visits per patient among the 5090 patients (minimum, maximum, median, percentiles 25 and 75, for example) and the interval between consecutive visits.

23. Line 200 – I don’t think the word “attrition” is the best choice here. It suggests losses to follow up in the study.

24. Table 1 – the difference may not be statistically significant, strangely, but from 60% to 81% insured is a large and very important difference. Its impact should be discussed. Statistical significance is not important here because this comparison of characteristics is not hypothesis testing. It is the magnitude of the difference that matters.

25. Lines 216-217 – it is very strange to register only systolic or only diastolic BP in so many patients. How do the authors explain this, taking into account their knowledge of local practice?

26. Line 218 – The 12821 patients with “at least 1 blood pressure measurement during the study period” include those with only systolic or only diastolic?

27. Line 221 – the association with number of visits is very likely to represent reverse causality. This limitation could be addressed by taking into account the reason for the visits.

28. Table 2 – the limits of the age categories are above or below? For example: 18-30, 30-45 – in which class are those aged 30?

29. Lines 236-238 – I suggest drawing and showing a Venn diagram with the 3 criteria, to show how many patients were common among them.

30. Line 244 – the logistic regression model in this study assesses association, not prediction.

31. Table 3 – the data shown are not trends. They are just numbers.

32. Table 3 – clonidine and alfuzosine are swapped (change categories)

33. Line 272 – how can untreated patients be controlled? Probably the problem is in the diagnosis as a case… The hundreds of papers on hypertension prevalence, awareness, treatment and control, estimate control among the treated or control among all hypertensives (defined and treated and controlled). It is not seen as defined here.

34. The association between treatment and control is weak (OR close to 2). Please discuss this more thoroughly.

35. Lines 327-328 – inability to afford medication is repeated

36. Lines 399-410 – this paragraph is too long for a conclusion. Please stick to the direct answer to the study objective, without discussing the results.

37. Figure 2 – The fourth bar title should read “At least 2 BP measurements”. At least means it could be more. The third bar should disappear, in my opinion (see comment 16, above). The fifth bar title should read “Criteria for hypertension”.

38. Figure 3 – Were age and number of visits included as continuous variables in the models? The effect of the number of visits is probably non-linear, so this option might not be the best.

6. PLOS authors have the option to publish the peer review history of their article (what does this mean?). If published, this will include your full peer review and any attached files.

**Do you want your identity to be public for this peer review?** For information about this choice, including consent withdrawal, please see our Privacy Policy.

Reviewer #1: No

Reviewer #2: **Yes: **Ana Azevedo

---

## [Decision Letter · Decision Letter 1]

6 Apr 2022

Hypertension screening, prevalence, treatment, and control at a large private hospital in Kampala, Uganda: a retrospective analysis

PGPH-D-21-00519R1

Dear Dr. Heller,

We are pleased to inform you that your manuscript 'Hypertension screening, prevalence, treatment, and control at a large private hospital in Kampala, Uganda: a retrospective analysis' has been provisionally accepted for publication in PLOS Global Public Health.

Best regards,

Andre F. S. Amaral, Ph.D.

Academic Editor

No further comments.

Reviewer Comments (if any, and for reference):

Reviewer's Responses to Questions

**Comments to the Author**

1. If the authors have adequately addressed your comments raised in a previous round of review and you feel that this manuscript is now acceptable for publication, you may indicate that here to bypass the “Comments to the Author” section, enter your conflict of interest statement in the “Confidential to Editor” section, and submit your "Accept" recommendation.

Reviewer #2: All comments have been addressed

2. Does this manuscript meet PLOS Global Public Health’s publication criteria? Is the manuscript technically sound, and do the data support the conclusions? The manuscript must describe methodologically and ethically rigorous research with conclusions that are appropriately drawn based on the data presented.

Reviewer #2: Yes

3. Has the statistical analysis been performed appropriately and rigorously?

Reviewer #2: Yes

4. Have the authors made all data underlying the findings in their manuscript fully available (please refer to the Data Availability Statement at the start of the manuscript PDF file)?

Reviewer #2: (No Response)

5. Is the manuscript presented in an intelligible fashion and written in standard English?

Reviewer #2: Yes

6. Review Comments to the Author

Reviewer #2: (No Response)

7. PLOS authors have the option to publish the peer review history of their article (what does this mean?). If published, this will include your full peer review and any attached files.

**Do you want your identity to be public for this peer review?** For information about this choice, including consent withdrawal, please see our Privacy Policy.

Reviewer #2: **Yes: **Ana Azevedo
